# Infantile Hemangioma: Risk Factors and Management in a Preterm Patient—A Case Report

**DOI:** 10.3390/reports7010003

**Published:** 2024-01-01

**Authors:** Florica Sandru, Aida Petca, Andreea-Maria Radu, Andrei Gheorghe Preda, Alina Turenschi, Andreea Teodora Constantin, Raluca-Gabriela Miulescu

**Affiliations:** 1Department of Dermatovenerology, University of Medicine and Pharmacy “Carol Davila”, 020021 Bucharest, Romania; 2Dermatology Department, Elias Emergency University Hospital, 011461 Bucharest, Romania; 3Department of Obstetrics and Gynecology, University of Medicine and Pharmacy “Carol Davila”, 050474 Bucharest, Romania; 4Department of Obstetrics and Gynecology, Elias Emergency University Hospital, 011461 Bucharest, Romania; 5Neonatology Department, Clinical Hospital of Obstetrics and Gynecology “Prof. Dr. P. Sârbu”, 060251 Bucharest, Romania; 6Pediatric Hospital Ploiesti, 100326 Ploiesti, Romania; alinaburcuta@yahoo.com (A.T.);; 7Department of Pediatrics, University of Medicine and Pharmacy “Carol Davila”, 020021 Bucharest, Romania; 8Pediatrics Department, National Institute for Mother and Child Health “Alessandrescu-Rusescu”, 20382 Bucharest, Romania; 9Department of Pharmacology, University of Medicine and Pharmacy “Carol Davila”, 020021 Bucharest, Romania

**Keywords:** hemangioma, premature, very low birth weight, propranolol, complications

## Abstract

Infantile hemangiomas (IHs), boasting a prevalence ranging from 4% to 10%, stand as the most commonly encountered benign tumors during the early stages of human life. We present the case of a 2-year-9-month-old child who was born preterm with very low birth weight (VLBW), 1010 g birth weight, at 27 weeks gestational age. During pregnancy, her mother had anemia and needed cervical cerclage. On her 10th day of life, the appearance of a frontal hemangioma could be observed. The hemangioma was situated at the hairline. At the age of one month, another hemangioma could be observed on her right arm. The hemangiomas were treated with propranolol oral suspension for 10 months and afterwards with local ointment for 2 months. This choice of treatment delivered great results, with no adverse reactions reported. In this case report, we underlined the risk factors for IH, possible complications, and available treatment options.

## 1. Introduction

Infantile hemangiomas (IHs), boasting a prevalence ranging from 4% to 10%, stand as the most commonly encountered benign tumors manifesting during the early stages of human life [1,2]. These lesions exhibit a distinct pattern of development, marked by a period of accelerated proliferation that is then followed by gradual regression. This particular feature of varying growth phases is the distinguishing characteristic that differentiates IH from vascular malformations [3,4,5]. Approximately one-third of hemangiomas are present at birth, while 40% become visible during the subsequent 4–6 weeks. The remaining one-third of hemangiomas typically manifest by the age of 6 months [6,7,8]. According to existing research, the proliferation period typically concludes by 9 months of age, whereas the following progressive reduction period may continue until 48 months of age [8,9].

Regarding their clinical manifestation, superficial hemangiomas may be seen as distinct pink-red macules, papules, or plaques, with their expansion being limited to the papillary dermis [10]. On the other hand, deep infantile hemangiomas occur as subcutaneous nodules with a light blue appearance [10]. The site of the lesions may manifest in several locations, both externally and internally, with the skin and soft tissue being the most prevalent sites [6,11]. As far as cutaneous localizations are concerned, the anterior cheek, pre-auricular region, and forehead are the most encountered, as approximately 80% of IHs occur on the facial and neck regions [6,12].

Various risk factors have been associated with the development of IH. These include female gender, low birth weight (LBW), placental anomalies, early pregnancy vaginal bleeding, use of in vitro fertilization, and a family history of hemangioma [3,13]. Preeclampsia, another instance in which LBW is encountered, can ultimately lead to HELLP syndrome, the latter being recognized as a risk factor for IH [14,15]. The global prevalence of obesity, particularly noteworthy at 38.3% among women in the United States, raises concerns. Special considerations must be taken regarding the heightened vulnerability of obese pregnant women to adverse pregnancy outcomes, including hypertensive disorders, cardiovascular diseases, and an elevated risk of preterm delivery. Despite these compelling factors, existing literature does not encompass studies associating obesity with the development of IH [16,17]. Moreover, in a study conducted in 2023 that analyzed the associated risk factors for developing IH, obesity, as well as the mode of delivery, did not have any statistically significant correlation with IH [18].

The importance of investigating patients with IH and treating this cutaneous disorder stems from the severity of syndromes that IH is associated with, such as PHACE syndrome, which refers to posterior fossa anomalies, hemangioma, arterial, cardiac, and eye anomalies, or LUMBAR syndrome, whose acronym stands for IH localized in the lower part of the body, urogenital anomalies, ulceration, myelopathy, deformity of the bones, anorectal, arterial, and renal anomalies [8,19]. Another factor that should be taken into consideration is represented by the complications that these benign tumors may impose, mainly ulceration, physical disfigurement, or functional impairment, depending on the localization of the IH [20,21,22].

Therapeutic actions are designed to be taken during the proliferative phase in order to prevent or reduce the occurrence of complications or to handle and alleviate the existing ones. However, systemic medical treatments should be considered to avert complications associated with both proliferative and involuting phases [23].

In the context of therapeutic interventions, it is noteworthy to mention that prior to the year 2008, the agents that were commonly advocated for utilization encompassed topical, intralesional, or systemic corticosteroids. Nevertheless, the harmful effects associated with their administration, including heightened susceptibility to infections, skin atrophy, decreased bone density, elevated blood sugar levels, elevated blood pressure, and restricted growth, impeded the administration of the therapeutic regimens [24,25]. Ever since the groundbreaking publication by Leaute-Labreze et al. elucidating the deeply beneficial therapeutic effects of propranolol on IHs, numerous subsequent investigations have been undertaken to explore the safety profile of β-blocker administration in neonates [26]. Currently, β-blockers serve as the primary treatment option in the management of IH [26].

In this case report, we present the case of a preterm-born infant who developed an IH 10 days after birth and a second one a month later. In our discussion, we address the risk factors affecting both the infant and the mother, explore associated comorbidities, examine potential complications, evaluate available treatment options, and analyze the overall patient outcomes. Moreover, we want to underline the anxiety that parents experience when physical disfigurement is included and the importance of early treatment in alleviating these fears.

## 2. Detailed Case Description

We present the case of a 2-year-9-month-old female who developed several infantile hemangiomas. During pregnancy, no significant data anomalies were observed, except for the noteworthy detail that the mother of the newborn experienced anemia and required cervical cerclage.

The patient was born through c-section at 27 weeks gestational age, thus weighing only 1010 g, a fact that situates her in the very low birth weight (VLBW) category of newborns. The newborn faced initial challenges, as reflected by an APGAR score of 5 at one minute, which improved to 6 at five minutes and 7 at ten minutes. She had pale-cyanotic skin and marked acrocyanosis; also, despite positive pressure ventilation and 50% oxygen supplementation, there was no improvement in oxygen saturation. The medical team escalated treatment to endotracheal intubation and mechanical ventilation due to persistently low oxygen levels.

Considering all the above, the newborn was admitted to the neonatal intensive care unit (NICU), where she was placed in a temperature-controlled environment. Intravenous caffeine treatment and fluids were initiated immediately after birth, and the patient received surfactant at a dosage of 200 mg/kg/day through the endotracheal tube. Subsequently, she underwent mechanical ventilation employing both high frequency oscillatory ventilation (HFOV) mode and synchronized intermittent mandatory ventilation (SIMV) mode for a duration of 22 days. Throughout this period, her oxygen saturation (SpO2) consistently remained above 95%. Peripheral cultures were also obtained at birth. Skin, stool, and nasal cultures were positive for Klebsiella pneumoniae, but the blood culture was fortunately negative. Intravenous Amikacin was added to the newborn’s treatment upon discovering the mother’s positive c-section scar culture for Klebsiella, which seemed susceptible to this antibiotic. Intravenous fluconazole was also administered for antifungal protection. Enteral nutrition began on the fifth day of life, initially through oral gavage, with gradual increases in formula quantity based on digestive tolerance and weight gain. The patient experienced jaundice with multiple etiology was successfully treated with phototherapy. Additionally, she had prematurity-related anemia, necessitating five blood transfusions for management. Due to the baby’s preterm birth, an ophthalmologic check-up was deemed necessary. The examination revealed an immature retina in the II area, posteriorly, with no identified Plus disease factors.

On the tenth day of her life, the appearance of a pink-red papule was observed in the frontal region, precisely at the site of the hairline. Due to its accelerated growth rate, the lesion was clinically diagnosed as a hemangioma (Figure 1). By the age of one month, another hemangioma had become apparent on her right arm. With an IHRes score of 5 points, due to the localization and size of the tumors, it was recommended that the patient receive care in a specialized center with expertise in infantile hemangioma management, such as our clinic. Due to the initial manifestation of the hemangioma on the highly vascularized scalp, posing an elevated risk of injury and subsequent bleeding for the infant, coupled with the tuberous nature and considerable dimensions of the hemangioma, indicating a reduced likelihood of responsiveness to sole topical treatment (such as timolol 10%), we made the decision to pursue systemic treatment as the primary therapeutic approach. Hence, the patient underwent a treatment regimen involving oral propranolol suspension for a duration of 10 months, followed by subsequent management with propranolol ointment for an additional 2 months.

Due to these cutaneous findings, several imaging investigations were conducted. The transfontanellar ultrasound revealed a subependymal cyst on the left side and a grade 2/3 intraventricular hemorrhage. The abdominal ultrasound showed no indications of intrahepatic or intrasplenic hemangioma. The cardiac ultrasound exhibited an atrial septal ostium secundum defect, a small patent ductus arteriosus, and a minor interatrial septum aneurysm. Despite the identification of these abnormal results, the patient did not meet the diagnostic criteria for either of the syndromes generally associated with infantile hemangiomas.

Upon discharge from our clinic, the infant exhibited a favorable general appearance, a healthy pink complexion, and the persistence of the two growing hemangiomas (frontal hemangioma measuring 2.5/2 cm, arm hemangioma measuring 0.7/0.5 cm) (Figure 2). She maintained stability from both cardiac and pulmonary perspectives, and her nutritional needs were met through bottle feeding, weighting 2290 g at discharge. Neurologically, she displayed good tonus and reactivity.

At 2 years of age, the patient underwent a follow-up check-up, during which a notable improvement was depicted in the clinical appearance of the hemangiomas. The tumor exhibited substantial reductions in both dimensions and vascularization following the treatment period. Furthermore, the patient remained free of any incidents of bleeding throughout the treatment, and notably, there were no reported adverse effects associated with the administration of propranolol. Hence, the efficacy of the aforementioned propranolol treatment was substantiated (Figure 3).

## 3. Discussion

Several risk factors have been statistically established as contributing to the development of IH. These include LBW, preterm birth, female sex, Caucasian race, progesterone therapies, multiple gestations, and a family history of IH. Nevertheless, existing research has put forward other risk factors, including IVF, advanced maternal age, placental anomalies, and maternal tobacco use [8,27]. In our case, the patient presented was a girl of Caucasian race, born preterm at 27 weeks gestational age, and had a VLBW, specifically 1010 g, consistent with the risk factors presented in the literature.

Local hypoxia is a significant element in the development of IH, as it leads to the suppression of the hypoxia-inducible factor pathway (HIF-pathway) and subsequent dysregulated proliferation [28,29]. The condition of anemia is often seen in pregnant women and is caused by several factors. Anemia occurring during pregnancy is often linked to a decrease in hemoglobin levels, hence rendering the fetus more susceptible to hypoxia [14]. The aforementioned condition has been demonstrated to exhibit a correlation with heightened occurrences of maternal and perinatal morbidities, such as LBW, preterm delivery, preeclampsia, and placenta previa [30]. Moreover, according to a research paper that included 1033 IH patients, anemia during pregnancy should be recognized as a significant independent risk factor for the development of IH, as it was present in a larger percentage of cases of IH compared to controls (25.46% vs. 6.10%, *p* < 0.0001) [14]. In the case we presented, the mother of the patient was diagnosed with anemia during pregnancy, making her and her infant susceptible to preterm delivery, LBW, and IH.

A study conducted in 2016 analyzed the inheritance pattern of IH. Among the 185 IH patients, one-third had a positive family history of IH, and 11 of them had more than four family members affected by IH. Moreover, in the majority of cases, they were a first-degree relative (65%) [31]. When compared to the sporadic cases, the characteristics of IH as well as the perinatal background did not differ [31]. According to this paper, the predominant inheritance pattern observed was autosomal dominant with incomplete penetrance [31]. Further genetic testing is required to determine the extent of inheritance patterns among IH patients and whether genetic counseling is warranted for these families.

As mentioned above, extensive IHs have been associated with multiple syndromes, such as PHACE and LUMBAR syndrome. PHACE, a syndrome whose acronym refers to posterior fossa malformation, hemangiomas, arterial, cardiac, and eye abnormalities, has been observed to be frequently linked with maternal preeclampsia or placental defects [20]. A prospective study evaluating the prevalence of PHACE syndrome among neonates with IH reported a rate of 2.3% of patients who were diagnosed with this syndrome [32]. his encountered in LUMBAR syndrome (lower body IH, urogenital abnormalities, ulceration, myelopathy, bony deformity, anorectal and arterial malformations, renal abnormalities) are frequently localized on the lower extremities of the body, specifically in areas like the lumbosacral and perineum [8,21,33,34,35]. Thus, our patient underwent several investigations, namely transfontanellar and abdominal ultrasound, cardiac ultrasound, as well as an ophtalmological examination. Even though several abnormal results were found (subependymal cyst, intraventricular hemorrhage grade 2/3, atrial septal ostium secundum defect, small patent ductus arteriosus, a small interatrial septum aneurysm, and immature retina in the II area, posteriorly), they did not meet the diagnostic criteria of either of the aforementioned syndromes.

As far as treatment is concerned, there is a limited number of indications for therapy during the proliferative period, which may be classified into two main categories: the first category is represented by measures aimed at preventing or minimizing the occurrence of morbidities, and the second category is comprised of strategies focused on managing and addressing existing morbidities [23]. Systemic medical treatments are recommended for the prevention of complications related to both proliferative and involuting-phase morbidities [23]. In the case where the IH is situated in a region of anatomical sensitivity, such as the facial region, or if there is an anticipation of rapid proliferation resulting in disfigurement, or if there are apprehensions regarding potential functional impairment if the growth of the IH is not addressed, it is recommended to pursue medical therapy [36]. The objective of medical therapy is to mitigate the growth of IH in order to reduce the likelihood of ulceration/bleeding and the potential deformity that may arise in the absence of intervention [23]. Insufficient empirical evidence exists for the establishment of a definitive threshold for commencing therapy with oral propranolol. However, a clinical trial published in 2022 evaluating factors that could lead to a greater likelihood of achieving positive outcomes from propranolol treatment showed that the commencement of therapy prior to 10 weeks of age had an elevated rate of treatment efficacy compared to the patients that received therapy after 10 weeks of age (86% vs. 60%) [37].

The research conducted by Leaute-Labreze et al. in 2008 marked a significant milestone in the management of IH. This study included the administration of systemic therapy using propranolol to two babies with IH who also had cardiac disease. Throughout the treatment period, a notable regression of the IH was noted [26]. Since then, multiple studies have been conducted to evaluate the safety and efficacy of β-blockers other than propranolol. A comprehensive multicenter, randomized clinical study was conducted to evaluate the effectiveness and safety of propranolol and atenolol in a cohort of 377 patients diagnosed with IH [38]. During the first assessment conducted after a period of 6 months, it was seen that the group administered with propranolol exhibited a higher response rate compared to the group administered with atenolol (93.7% vs. 92.5%) [38]. Nevertheless, over a span of two years, both cohorts exhibited a comparable rate of response to the prescribed intervention. Furthermore, the propranolol group had a higher incidence of side effects. The findings of this clinical research indicate that atenolol is a viable and secure choice for the treatment of troublesome IH, whether as the primary therapeutic option or as an alternative for individuals who are unable to use propranolol due to contraindications or intolerance [38]. In a clinical trial published in 2022, nadolol also proved to be a viable option for IH treatment, delivering a faster and enhanced response in treating IH [39]. However, additional studies are required to establish its superiority in comparison to propranolol [39]. Other available therapies for IHs are corticosteroids, laser, or surgical therapies, both in the proliferative and involuting phases of IH. However, in the involuted IH disfigurement, surgery represents the key modality to obtain an aesthetic result by removing the inelastic skin, healing the ulceration scar, and recountouring the damaged area [23]. In a recent review published in 2023, β-blockers were identified as a successful initial approach specifically for superficial infantile hemangiomas. On the other hand, the pulsed dye laser procedure has proven effective for treating deep hemangiomas, expediting the involution process when used in conjunction with other treatment modalities, and reducing scarring in ulcerated hemangiomas [40].

In our case, the patient received treatment with propranolol oral suspension for 10 months, followed by a period of local treatment with ointment for 2 months. This choice of treatment delivered great results (Figure 3), with no adverse reactions reported. Moreover, for the residual lesions of IH after propranolol treatment, the patient may undergo Nd:YAG 1064 nm laser, which, according to a prospective cohort study, should be taken into consideration as a second-line therapy for the remaining lesions of IHs [41].

## 4. Conclusions

Drawing upon a comprehensive examination of the case, the patient presented had multiple risk factors for developing IH, namely: premature birth, VLBW, Caucasian race, female sex, as well as maternal anemia during pregnancy. The therapeutic approach for IH included oral propranolol suspension for 10 months, followed by propranolol ointment for 2 months, with remarkable improvement in the clinical aspect of the lesions. Moreover, by presenting this case, we underlined the importance of early therapy administration in IHs in preventing complications and obtaining remarkably aesthetic results in the clinical appearance of the lesions.

## Figures and Tables

**Figure 1 reports-07-00003-f001:**
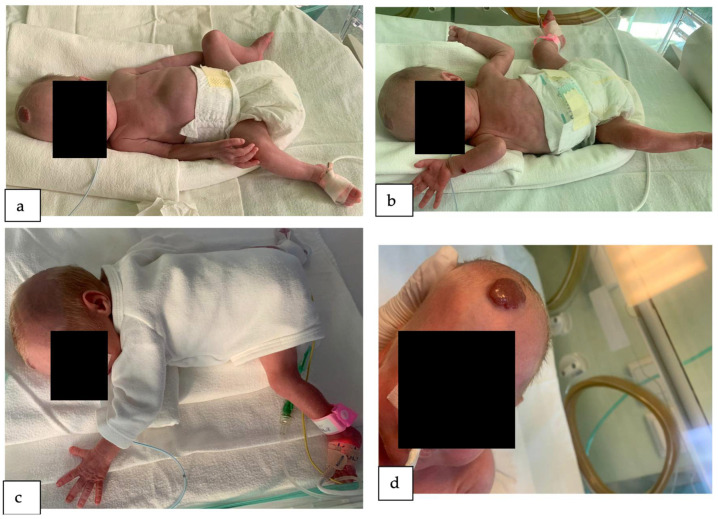
The infant in the intensive care unit. (**a**) The infant 30 days old. Notice the hemangioma on her forehead. The infant was being fed through gavage. (**b**) The infant is 40 days old. The hemangioma on the forehead increased its dimensions. Also notice the hemangioma on her right forearm. (**c**) the infant 45 days old. (**d**) the infant 50 days old. Close-up of the hemangioma on her forehead.

**Figure 2 reports-07-00003-f002:**
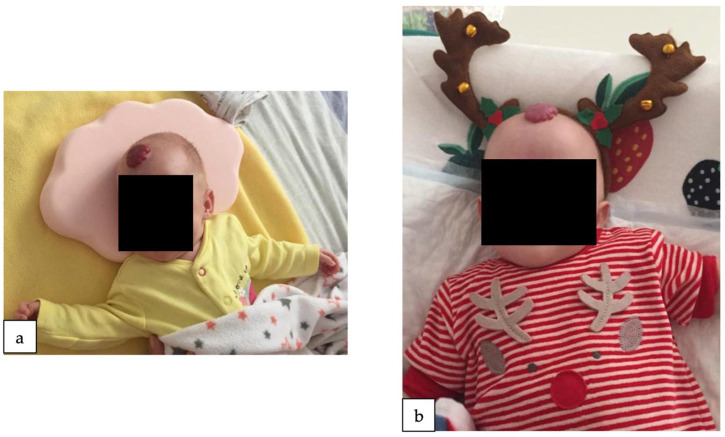
The infant at 2 months, 3 weeks (**a**), and 4 months (**b**). Notice the impressive forehead hemangioma.

**Figure 3 reports-07-00003-f003:**
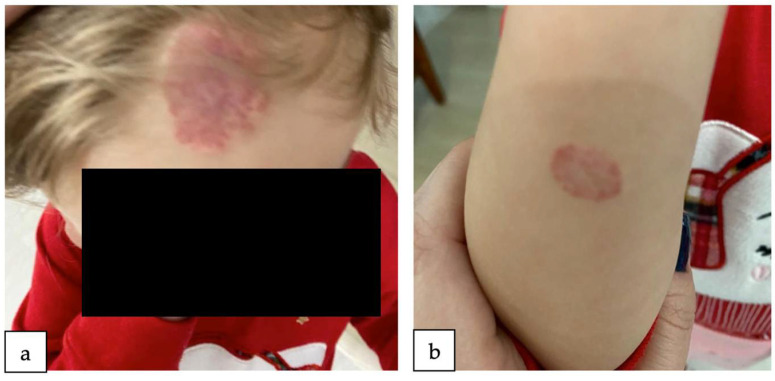
The patient, 2 years old. Notice the improvement of the hemangioma on her forehead (**a**) and forearm (**b**). The hemangiomas were treated with propranolol oral suspension for 10 months and afterwards with local ointment for 2 months.

## Data Availability

The data presented in this study are available on request from the corresponding author. The data are not publicly available due to privacy.

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
