# Peer review of "Infantile Hemangioma: Risk Factors and Management in a Preterm Patient—A Case Report"

_reports, 2024, doi:10.3390/reports7010003_

Round 1

Reviewer 1 Report

Comments and Suggestions for Authors

The authors presented a very well-documented case of infantile hemangioma of a 2-year-old and 9-month-old child. For this case, both maternal risk factors (e.g. maternal anemia) and neonatal risk factors were analytically presented: female gender, prematurity, VLBW, Apgar score 5 at one minute and 7 at 10 minutes, need for resuscitation, mechanical ventilation etc., regarding the occurrence of infantile hemangioma.

At the same time, in this case presentation, the treatment and evolution of the case was described in detail, from birth to the age of 2 years and 9 months.

In section 3. Discussion, the authors present the risk factors for infantile hemangioma mentioned in the specialized literature including: maternal and perinatal morbidities (LBW, premature birth, preeclampsia and placenta previa).

Following the analysis of the presented case and in correlation with the data from the specialized literature, the authors direct their attention particularly to maternal anemia as a risk factor for IH and fully document this aspect.

The results presented by the authors complement and bring very important clarifications regarding the risk factors of infantile hemangioma and its therapeutic management.

I recommend rewording the conclusions to clearly highlight the results of the analysis of the presented case.

In their current form, the conclusions present only a few general aspects.

Reviewer 2 Report

Comments and Suggestions for Authors

The manuscript explored the treatment of infantile hemangioma in a case of a child born prematurely. My overall evaluation of the manuscript is positive. There are a number of minor revisions, formal and scientific aspects that should be addressed.

1.Regarding the treatment method, why did they not use Timolol?

2. Regarding the cause of the disease, it is necessary to conduct a more detailed investigation based on statistics. In addition, if the cause of genetics is mentioned, it is necessary to suggest to the parents that genetic tests should be done before pregnancy.

3. The results are not clearly described. Therefore, it is necessary to rewrite the result part.

Reviewer 3 Report

Comments and Suggestions for Authors

The authors present a case of a prematurely born girl with two infantile hemangiomas which were treated with propranolol in the first year of life. Although the message of the paper is clear (early initiation of treatment, exclusion of more dangerous pathology such as PHACE or LUMBAR syndromes) it includes a lot of redundant information that is completely irrelevant for the purpose of the article (such as lines 93 - 106, 2018 - 223).

Some approaches used are even difficult to understand - such as Meropenem and Linezolid as empirical antibiotic therapy.

Use of the term "cavernous" hemangioma is also controversial as this diagnosis describes a venous malformation (so it is not synonymous with infantile hemangioma).

Some parts of Discussion should be at least partly included in Introduction (lines 236 - 245, 252 - 270).

References include some quite old publications (2, 6, 7, 8, 9, 10, 19, 20), but some more recent review articles are missing, such as Leung AKC et al (An updated review in Curr Pediatr Rev 2021) or Dahan E (A review of current treatment options in Pediatr Ann 2023).

Comments on the Quality of English Language

In addition to typo corrections, extensive English corrections are also necessary.

Round 2

Reviewer 2 Report

Comments and Suggestions for Authors

Dear Editor

The article is acceptable with the given changes.

Author Response

Thank you!

Reviewer 3 Report

Comments and Suggestions for Authors

The authors took into account all comments and supplemented the text of the paper accordingly. 

Although I am not a native English speaker (despite some of the corrections made), I still think that the text should be checked by a proofreader.

Author Response

Dear Reviewer,

I, Dr. Andreea Maria Radu, one of the authors of the paper submitted, wish to affirm my possession of the Cambridge Certificate of Proficiency in English (C2 level). I have thoroughly reviewed and assessed the quality of the English language in the final manuscript, and I find it suitable for publication. Moreover, my colleague, Alexandra Maria Roman, who was approached by us, the authors, to undertake a language assessment of the document also found that the paper is suitable for publication. She also possesses the Cambridge Certificate of Proficiency in English (C2 level).

However, we did find small typos when we reviewed the paper again, which you will find highlighted in yellow.

We attached our Cambridge Certificates in the supplementary files.

Thank you!